# The Influence of Different SARS-CoV-2 Strains on Changes in Maximal Oxygen Consumption, Ventilatory Efficiency and Oxygen Pulse of Elite Athletes

**DOI:** 10.3390/diagnostics13091574

**Published:** 2023-04-27

**Authors:** Dragutin Stojmenovic, Tamara Stojmenovic, Marija Andjelkovic, Nenad Trunic, Nenad Dikic, Natasa Kilibarda, Ivan Nikolic, Ivana Nedeljkovic, Marina Ostojic, Milos Purkovic, Jovana Radovanovic

**Affiliations:** 1Department of Physiology, Faculty of Medical Sciences, University of Kragujevac, 34000 Kragujevac, Serbia; dragutin.stojmenovic@gmail.com (D.S.); nikolic.ivan6@gmail.com (I.N.); purkovic85@gmail.com (M.P.); djukanoviceva@yahoo.com (J.R.); 2Faculty of Physical Education and Sports Management, University of Singidunum, 11000 Belgrade, Serbia; ntrunic@singidunum.ac.rs (N.T.); ndikic@singidunum.ac.rs (N.D.); 3Department of Pharmacy, University of Singidunum, 11000 Belgrade, Serbia; mandjelkovic@singidunum.ac.rs (M.A.); nkilibarda@singidunum.ac.rs (N.K.); 4Cardiology Department, Clinical Center of Serbia, Faculty of Medicine, University of Belgrade, 11000 Belgrade, Serbia; ivannanedeljkovic@yahoo.com (I.N.); drmarinaostojic@gmail.com (M.O.)

**Keywords:** SARS-CoV-2, functional capacity, professional athletes, respiration, aerobic endurance

## Abstract

Purpose: The aim of this study was to evaluate the influence of different SARS-CoV-2 strains on the functional capacity of athletes. Methods: In total, 220 athletes underwent cardiopulmonary exercise testing (CPET) after coronavirus infection and before returning to sports activities. Eighty-eight athletes were infected by the Wuhan virus, and 66 were infected during the Delta and Omicron strain periods of the pandemic. Results: The CPET results showed significantly decreased maximal oxygen consumption, ventilatory efficiency, and oxygen pulse in athletes who were infected with Wuhan and Delta strains compared to athletes who suffered from Omicron virus infection. An early transition from aerobic to anaerobic metabolic pathways for energy production was observed in the Wuhan and Delta groups but not in athletes who were infected with the Omicron strain. There were no differences in the obtained results when Wuhan and Delta virus variants were compared. Conclusion: These results suggest that the Wuhan and Delta virus strains had a significantly greater negative impact on the functional abilities of athletes compared to the Omicron virus variant, especially in terms of aerobic capacity and cardiorespiratory function.

## 1. Introduction

The pandemic of the SARS-CoV-2 virus has left enormous consequences for the lives and health of ordinary people, as well as athletes [1]. The virus has created a major health problem worldwide, and the most common symptoms initially reported were respiratory and cardiovascular [2], but SARS-CoV-2 infection was later shown to be a multisystemic disease [3]. Since 2020, the coronavirus pandemic has changed the entire organization of all sports events, including training sessions themselves, especially when it comes to professional sports [4]. After the lockdown period and the two-month home isolation in 2020, athletes returned to training and competitions, but under different circumstances, in closed balloons and with constant polymerase chain reaction (PCR) testing for the virus. Quarantine itself left its mark on their fitness levels because, except for at home, they did not have the opportunity to train or compete [5]. 

At first, it was assumed that older people and those with underlying medical conditions were more likely to become infected and develop serious illnesses. Considering that athletes belong to a young and healthy population, at the beginning of the pandemic, the recommendations for returning to sports activities included only a physical examination and electrocardiogram (ECG) in rest, with basic laboratory analyses, following a minimum of fourteen days of home isolation [6,7]. Over time, it became clear that even young, healthy, and physically active individuals could end up with clinically serious symptoms and life-threatening forms of the disease. Frequent cases of myocarditis, pericarditis, exercise intolerance, and dyspnea on exertion, even after mild forms of the infection, indicated the need for more detailed diagnostic procedures, such as more extensive laboratory analyses (inflammatory and cardiac biomarkers), echocardiography, and cardiopulmonary exercise testing [7,8,9,10,11]. More detailed medical examinations were necessary to preserve the athlete’s health and avoid the possibility of sudden cardiac death.

Since the beginning of the pandemic in March 2020, the SARS-CoV-2 virus has mutated over time, resulting in genetic variations in the population of circulating viral strains. These mutations may impact virus transmission or the severity of symptoms in infected individuals [12]. According to the official data of the national body for infectious diseases, in the period between 2020 and 2022, three dominant strains of SARS-CoV-2 virus were recorded in Serbia: the Wuhan, Delta, and Omicron strains [13]. Scientific research on the topic of various strains of the virus is primarily based on the impact of mutations in the SARS CoV-2 spike on viral infectivity and antigenicity [14,15]. On the other hand, some studies have investigated the effects of the vaccine against new variants of the virus [16,17]. There is currently a lack of scientific research regarding the impact of a particular strain of coronavirus on the severity of infection, clinical symptoms, and the development of cardiorespiratory and other forms of diseases, both in the general population and in athletes. With regard to athletes, the latest studies indicate a decline in functional abilities after SARS-CoV-2 infection, with increased respiratory and metabolic demands [18,19,20]. Furthermore, different forms of inflammatory heart diseases (myocarditis and pericarditis), decreased lung capacity, and aerobic endurance have been reported as a result of SARS-CoV-2 infection [9,21,22,23,24]. However, there is no research regarding the impact of different virus strains on these parameters. Even though it is now known that SARS-CoV-2 infection can impair athletes’ health and sports performance, it remains unclear to what extent different strains of the coronavirus affect their functional ability. The question arises whether different variants of the virus affect the cardiorespiratory fitness of athletes in the same way or whether the potential of the virus to impair these abilities is slowly weakening as the pandemic continues. Considering all of the above, the aim of this study was to evaluate the impact of different strains of the coronavirus on the overall functional capacity of professional athletes during the ongoing pandemic.

## 2. Material and Methods

### 2.1. Participants and Study Design

Two hundred and twenty (N = 220) elite athletes from Serbia participated in this prospective cohort study. In total, 112 soccer players (age 23.05 + 4.64) from five professional senior Serbian teams (The Serbian First League; UEFA Europa League) and 108 basketball players (age 24.52 + 4.80) from six professional Serbian teams (Euroleague and ABA League) were involved in the study. This research was conducted during the period from September 2020 to September 2022. The criteria for inclusion in the study were prior SARS-CoV-2 infection, which had been confirmed by a polymerase chain reaction test. There were two indications for conducting the PCR tests. The first referred to athletes who had clear symptoms of infection. Another indication for conducting the PCR test was the mandatory testing of athletes before every official match, according to the proposals of domestic and European competitions. The Genome Sequencing Center within the Institute of Molecular Genetics and Genetic Engineering of Serbia conducted sequencing of the coronavirus genome to identify new coronavirus strains. Genome sequencing was carried out when a new strain was suspected and continued until the point when it was evident that the dominance of a new variant of the virus had emerged. Accordingly, study subjects were divided into three groups depending on the period of the pandemic during which they had been infected by SARS-CoV-2 and the dominance of one of the three strains of the virus in that time interval (the Wuhan, Delta, or Omicron strain).

The first group of participants consisted of athletes who were infected with the SARS-CoV-2 virus in the period between September 2020 and July 2021 (*n* = 88). Genome sequencing at the beginning of this period indicated the dominance of the Wuhan strain. The second group of athletes were subjects who were thought to have been predominantly infected with the Delta strain (*n* = 66). These participants had PCR positive tests in the period from August 2021 to January 2022. The third research group consisted of athletes tested from February 2022 to September 2022 (*n* = 66). By sequencing the genome of the virus during this period of the pandemic, it was concluded that the Omicron strain was dominant. Table 1 summarizes group criteria selection for participation in this research. 

All study participants reported asymptomatic or mild to moderate forms of infection. Athletes with symptomatic complaints mostly reported fever, mild shortness of breath, weakness, headache, ageusia, and anosmia. None of the subjects were hospitalized. They were home treated and kept in isolation for 14 to 30 days (an average of 22.3 days). 

After the cessation of symptoms of SARS-CoV-2 infection and/or a negative control PCR test, study participants underwent medical examinations with the aim of deciding on their capacity to return to sports activities. The medical examination included an electrocardiographic examination at rest, blood pressure measurement, and auscultation of the heart and lungs (physical examination). Additionally, laboratory analyses were performed with the aim of evaluating inflammatory and cardiac biomarkers, such as C-reactive protein (CRP), D-Dimer, NT-proB-type Natriuretic Peptide (nt-pro BNP), and high-sensitivity cardiac troponin T (hs-cTnT). Along with laboratory analyses, transthoracic 2D echocardiography was done to ensure that there was no ongoing acute inflammatory process and/or underlaying myo/pericarditis as absolute contraindications for performing cardiopulmonary exercise testing (CPET), which was the final medical exam prior to the decision of whether or not to return to play. 

CPET, as a maximal symptom-limited exercise test, was performed to evaluate the health status and functional capacity of athletes. Maximal exercise testing was performed on a treadmill. Subjects were equipped with a facemask, heart rate monitor (COSMED Wireless HR Monitor, Rome, Italy), and portable ECG device (Quarck T 12x, Wireless 12-lead ECG, Rome, Italy) to perform the test. According to the protocol for professional athletes, the initial speed and inclination were set at 6 km/h and 3°, respectively. Every 40 s, the treadmill speed was increased by 1 km/h, while the inclination remained constant throughout the test. Oxygen consumption kinetics were measured continuously using a breath-by-breath analysis technique (Quark CPET system and Omnia software manufactured by Cosmed, Rome, Italy). Heart rate was monitored by a portable ECG device. A test was considered maximal if participants achieved 90% or more of predicted maximal heart rate for age and gender (220–age), a plateau in oxygen consumption despite increased workload (plateau < 150 mL O_2_/min), and a respiratory exchange ratio greater than 1.10, together with reached volitional exhaustion. All tests were performed by medical doctors, and the test equipment was routinely calibrated with both volume and gas calibration before each testing procedure. 

### 2.2. Electrocardiographic Monitoring and Respiratory Function

Continuous ECG monitoring with 12-lead Stress ECG was performed to detect possible rhythm and conduction disturbances, as well as changes in the ST-T segment. The same device was used to obtain the maximal heart rate and assess the three-minute heart rate recovery after the maximal exhaustion test (Figure 1). Oxygen pulse (O_2_/HR), as an indirect indicator of left ventricular function (the volume of oxygen ejected from the ventricles with each cardiac contraction), was measured and assessed by a Wasserman 9-Panel Plot (Figure 2, Panel 2). In addition to the evaluation of the maximum value of the oxygen pulse at the end of the test, the kinetics of the O_2_/HR curve during CPET was monitored all the time with the aim of evaluating the contractility of the left ventricle in terms of meeting the body’s metabolic needs for oxygen. With an increase in heart rate and intensity of effort, the exponential growth of the O_2_/HR curve was expected to be a normal response during the test. The plateau of the curve growth occurred in the final stages of the CPET, at maximum intensity.

A Wasserman 9-Panel Plot was used to monitor the response of ventilatory equivalents for oxygen and carbon dioxide during CPET (VE/VO_2_ and VE/VCO_2_). The efficiency of the ventilatory pump at various workloads was continuously evaluated using Panel 4 of the Wasserman 9-Panel Plot (Figure 2). Overall ventilatory efficiency was calculated by the VE/VCO_2_ index using the breath-by-breath analysis technique at the end of the test. This was done by excluding data points after the onset of maximal hyperventilation at the maximal effort. The VE/VCO_2_ slope, as the relationship between minute ventilation and carbon dioxide production, is usually a hallmark characteristic of pulmonary vascular diseases or exercise intolerance and disability, which is why it was evaluated by Omnia software to assess lung function and exercise tolerance. 

### 2.3. Aerobic Capacity and Metabolic Response to Effort

Maximal oxygen consumption (VO_2_ max), as an objective and accurate indicator of cardiorespiratory fitness and aerobic endurance, was evaluated at the end of the CPET. The plateau in oxygen consumption was considered when determining the final VO_2_ max value (Figure 3). The oxygen consumption at the first ventilatory anaerobic threshold was obtained to evaluate aerobic economy, which is a measure of energy utilization when running at an aerobic intensity. 

The heart rate was obtained at the first ventilatory anaerobic threshold (VAT) and second ventilatory anaerobic threshold (or respiratory compensation point (RCP)) to determine the intensity of effort at which the transition from aerobic to anaerobic energy sources occurs. VAT and RCP were measured by using the “thresholds” panels of the Wasserman 9-Panel Plot (Panels 4, 7, and 8 in Figure 2 and Figure 4). The V-slope method and VE/VO_2_ curve kinetics were the methods of choice to obtain VAT. The V-slope method was used to visually determine the first point of departure from linearity of carbon dioxide output plotted against oxygen uptake. A sudden and continuous increase in the ventilatory equivalent for oxygen (VE/VO_2_) was also a sign of reaching the first threshold. Respiratory exchange ratio (RER), as a quotient of metabolic production of carbon dioxide and the uptake of oxygen (CO_2_/O_2_) was used to evaluate the point of RCP, or second ventilatory anaerobic threshold. The onset of absolute anaerobic metabolism was determined by measuring the heart rate at RER = 1. Furthermore, the simultaneous sudden increase in ventilatory equivalents for oxygen and carbon dioxide, as well as the sudden drop in the end-tidal partial pressure for carbon dioxide (PetCO_2_) indicated reaching the second ventilatory threshold. At this point, ventilatory requirements for delivering oxygen to the muscle cells and the removal of carbon dioxide into the external environment are extremely high. Additionally, the RER value was calculated at the end of the test (maximal value) to assess the metabolic response to the maximal effort and level of achieved anaerobic exertion. Figure 4 shows all the methods described above for determining thresholds.

### 2.4. Statistical Analysis

To describe parameters of importance, depending on their nature, the following were used: frequency, percentages, sample mean, sample median, sample standard deviation, rank, and 95% confidence intervals. To test the normality of the distribution, Shapiro–Wilk tests were used, as were graphs, both histogram and normal KK plot. To test the differences in functional parameters between athletes infected with different strains, analysis of variance (one-way ANOVA) was used, as was the Post Hoc Bonferroni test for multiple comparisons. Statistical data processing was performed using the statistical package SPSS 20.0 for Windows. Differences were considered significant when the *p* value was less than 0.05.

## 3. Results

Table 2 shows the results of the medical examination at rest. Heart rate at rest, blood pressure values, and average values of echocardiographic parameters were within normal limits for age, gender, and sports discipline. 

The results of laboratory analyses that were a prerequisite for conducting CPET are shown in Table 3. The values of inflammatory and cardiac biomarkers were within the normal range for all three groups of participants. 

Descriptive statistics for the obtained CPET variables are shown in Table 4.

The one-way ANOVA test showed statistically significantly higher VO_2_ max values in athletes who were infected with the Omicron virus variant compared to those who had been infected by the coronavirus disease during the dominance of the Wuhan and Delta strains (*p* < 0.01). This result was also confirmed by the Post Hoc Bonferroni test (*p* < 0.01). When we compared aerobic capacity between the Wuhan and Delta groups of participants, there was no statistically significant difference. Despite this fact, Figure 5 shows that a larger number of athletes, those who were infected with the Wuhan strain, had a lower VO_2_ max than the mean calculated value and compared to the same data within the Delta group. 

Oxygen consumption at VAT was also much greater in the Omicron group of participants compared to both the Wuhan and Delta groups, which means that the aerobic economy has improved to a great extent during the later stages of the pandemic. Furthermore, athletes infected with the Omicron virus variant had much better ventilatory efficiency and higher O_2_/HR values than those infected with the Wuhan and Delta strains (Figure 1 and Figure 6). VE/VCO_2_ slope values were significantly lower in the later stages of the pandemic, which indicates that the ventilatory requirements for a certain level of effort were reduced. The amount of oxygen delivered to the working muscles with each left ventricular contraction was highest in the Omicron group. The distribution of these data within different groups of SARS-CoV-2 strains is shown in Figure 7 and Figure 8, respectively.

Moreover, an early transition from aerobic to anaerobic metabolic pathways for adenosine triphosphate (ATP) production was absent during CPET in the Omicron group when compared to the Wuhan and Delta strains. Achieved heart rate values at VAT and RCP were much higher in these participants compared to the athletes from the Wuhan and Delta groups. On the other hand, there was no statistically significant difference either in terms of maximal achieved heart rate at the end of CPET or in terms of the three-minute heart rate recovery among all three study groups. The heart rate analysis and the influence of different SARS-CoV-2 strains on this data, during the different phases of the maximum effort test, are shown in Figure 2.

The one-way ANOVA and Post Hoc Bonferroni test confirmed a statistically significant difference between the achieved RER values at the end of the test when we compared Wuhan and Omicron and Delta and Omicron strains (*p* < 0.01). Exposure to early and pronounced metabolic fatigue during the period of dominance of the Wuhan and Delta strains was recorded by significantly high maximum RER values at the end of the CPET tests when compared to the Omicron group of athletes. Table 5 shows the connection between higher RER values and lower ventilatory thresholds. The lower the heart rates were on VAT and RCP, the higher the RER values achieved at the end of the CPET. In the Omicron group of study participants, the transition to anaerobic glycogenolysis occurred at a higher intensity of effort. This means that oxygen, as a source for creating ATP, was used longer during CPET, which lowered the final RER values and the level of anaerobic fatigue.

There was no statistically significant difference between athletes who were infected with the Wuhan and Delta variants of SARS-CoV-2 virus when considering all the evaluated variables (*p* > 0.05). During CPET, no malignant disorders of rhythm, conduction, or pathological changes of the ST-T segment were observed in any of the examined athletes. All athletes received permission to return to sports activities with a gradual entry into the training process, especially in the early stages of the pandemic, given that their functional capacity was significantly compromised.

## 4. Discussion

The results showed significantly decreased aerobic capacity of all tested athletes. Maximal oxygen consumption, as a measure of cardiorespiratory fitness, was much lower in the participants compared to the values anticipated for professional soccer and basketball players. Soccer players possess excellent endurance, with VO_2_ max reported to range between 55 and 70 mL/kg/min in elite performers [25]. On the other hand, maximal oxygen consumption in basketball players is, according to some authors, in the range of 42 to 59 mL/kg/min, while some authors in recent research showed VO_2_ max values in this type of sport to be between 45 and 65 mL/kg/min [26]. In general, according to the American College of Sports Medicine, VO_2_ max above 50 mL/kg/min is a desirable value for professional athletes who compete in ball sports. According to our results, a drop in VO_2_ max values was evident even though participants of our study were members of professional teams who competed at high levels of competition, both in domestic and European leagues. It is important to emphasize that most of the performed CPET tests were carried out during the competitive part of the season, when it is expected for players to be in peak-level condition. These results coincide with the results of most previous studies dealing with this topic, which showed that there is a decrease in VO_2_ max values in athletes who have overcome SARS-CoV-2 infection [18,19,23]. Even though VO_2_ max values were lower than expected for athletes in all three groups (Wuhan, Delta, and Omicron), there was an obvious increase in both aerobic capacity and aerobic economy as the pandemic progressed. In other words, athletes who were infected with the Omicron strain had a significantly higher VO_2_ max and expended oxygen more economically, delaying the early onset of anaerobic metabolism and fatigue. 

Even though ventilatory efficiency was within normal values (VE/VCO_2_ between 20 and 30) [27] in all three study groups, it was observed that athletes who were infected with the Omicron virus variant had decreased ventilatory requirements for a given level of exercise when compared to Wuhan and Delta participants. Oxygen delivery to the muscle cells and carbon dioxide removal were significantly more efficient in the Omicron patients, which explains the significantly higher VO_2_ max values in these athletes. Furthermore, O_2_/HR values were also within normal values for professional athletes (>20 mL/beat) throughout the pandemic [28], but significantly higher values of the oxygen pulse were observed in athletes who had had the Omicron variant infection. A more efficient ejection fraction of the left ventricle provides a greater amount of oxygen to the muscle cells and thus more energy for work. Furthermore, athletes who had suffered from a coronavirus infection during the periods of Wuhan and Delta strain dominance had an early transition from aerobic to anaerobic metabolic pathways for obtaining ATP. Thresholds were reached at significantly lower heart rates and levels of physical intensity compared to the athletes in the Omicron group. This could explain the much higher maximal RER values and prolonged anaerobic fatigue during CPET. 

Hypoxemia-associated changes in external respiration are the first objective indicators of the clinical signs of respiratory failure after SARS-CoV-2 infection [29]. Given that ventilatory efficiency and O_2_/HR were within normal limits, the coronavirus in all three groups of subjects did not impair external respiration, lung, or cardiovascular function. Therefore, low values of aerobic capacity and aerobic efficiency in all three groups of subjects, especially in athletes who suffered from Wuhan and Delta strain infection, can probably be explained by impaired internal respiration. Oxygen delivery to muscle cells was apparently adequate, but tissue utilization of oxygen was compromised. This could potentially be explained by the reduced number of mitochondria at the level of muscle cells and, thus, the inability to adequately utilize oxygen to produce ATP. The latest research related to pandemics indicates the potential of the coronavirus to damage mitochondria as the main oxygen organelles [30,31]. The reduced number of mitochondria at the level of muscle cells can thus explain the impaired oxidative phosphorylation at the cell level.

Additionally, only a few studies have dealt with the impact of different virus strains on the relevant clinical parameters in SARS-CoV-2 patients. The results of one such study showed that the Wuhan strain had a less favorable effect on the erythropoiesis process and led to more pronounced hypoxia in SARS-CoV-2 patients compared to other variants of this virus in later phases of the pandemic [32]. Considering this fact, the significantly reduced functional capacity in the Wuhan group of athletes, compared to those infected with Omicron virus variants in our study, could be explained by the lack of oxygen “carriers” through circulation. Some studies have shown that all the variants of concern (one being linked to rapid spread in human populations) manifested varied immune escape, especially the Omicron [15,17]. Although it has been proven that the Omicron strain has a high transmission potential, according to our results, its potential to impair an athlete’s functional abilities is obviously weaker. Much higher VO_2_ max VE/VCO_2_ slope, and O_2_/HR values, together with delayed transition from aerobic to anaerobic metabolism, were observed in the Omicron group of athletes when compared to other evaluated strains. 

## 5. Conclusions

The main results of the study indicate that the potential of the virus to impair an athlete’s functional abilities decreases as the pandemic progresses, which greatly facilitates a safe return-to-play decisions for both sports physicians and coaches. Although no pathological changes were observed after the SARS-CoV-2 infection, the virus overall reduced the functional capacity of the athletes, especially in the early stages of the pandemic during Wuhan and Delta strain dominance. This suggests that a gradual and cautious return to sports activities is still necessary to avoid exacerbation of potential underlying accute myo/pericarditis or lung disease.

## Data Availability

All data, stored anonymously, are available to all those who explicitly request it from the corresponding author.

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
