# Peer review of "The Influence of Different SARS-CoV-2 Strains on Changes in Maximal Oxygen Consumption, Ventilatory Efficiency and Oxygen Pulse of Elite Athletes"

_diagnostics, 2023, doi:10.3390/diagnostics13091574_

Round 1

Reviewer 1 Report

The aim of the study is to evaluate the influence of different SARS CoV-2 strains on functional capacity of athletes. The author concluded that SARS Cov-2 infection has reduced the functional capacity of the athletes and they suggest a gradual and cautious return to sports activities.

The results and the study conclusion are interesting but quite obvious. In my opinion the really add value of this article is the comparison between different type of SARS Cov-2.

I suggest to improve the conclusion of the abstract because they are too generic.

I suggest to reduce the introduction section.

I suggest to improve the method section: is the study retrospective or prospective? For the first paragraph may be useful to add a table to summarize the modality of selection because this part is not easy to read.

In the tables I suggest to add the p value.

In the picture I suggest to cut of some useless part to increase the dimension of the pictures.

Author Response

Dear reviewer,

First of all, thank you for taking the time to review this paper. We appreciate your positive and encouraging comments. Thank you very much for your insights which greatly improved the quality of our work. Below are the responses to your reviews:

Point 1: The results and the study conclusion are interesting but quite obvious. In my opinion the really add value of this article is the comparison between different type of SARS Cov-2.

Response 1: The conclusion of the paper has been modified in terms of first emphasizing the results related to the impact of the different strains of the virus on the functional abilities of athletes, and then we stated that the pandemic itself had an overall effect on the decline in the functional abilities of athletes, regardless of the strain of the virus.

Point 2: I suggest to improve the conclusion of the abstract because they are too generic.

Response 2: We changed the conclusion of the abstract in order to be less generic and to better emphasize the study results. 

Point 3: I suggest to reduce the introduction section.

Response 3: We reduced the introduction section as much as we could considering that we had to be careful about the number of words (3500-4000) in the article that was requested by the editors.

Point 4: I suggest to improve the method section: is the study retrospective or prospective? For the first paragraph may be useful to add a table to summarize the modality of selection because this part is not easy to read.

Response 4: In the method section, we have added the type of study that was conducted, and as you advised, we have added a table in order to summarize the modality of selection to make it clearer for interpretation.

Point 5: In the tables I suggest to add the p value.

Response 5: We added p values in the tables.

Point 6: In the picture I suggest to cut of some useless part to increase the dimension of the pictures.

Response 6: All unnecessary parts of the images are cropped in order to reduce the size of the pictures and improve their visibility.

Reviewer 2 Report

I have received for review an original research article entitled “The influence of different SARS CoV-2 strains on changes in maximal oxygen consumption, ventilatory efficiency and oxygen pulse of elite athletes” prepared by Dragutin Stojmenovic et al., which is being processed for publication in the journal Diagnostics (IF=3.992). The pandemic caused by the SARS-CoV-2 virus has been, and in some countries still is, a major public health challenge. Therefore, conducting scientific research that will allow us to better understand the nature of this virus and the clinical course of the disease it causes is extremely important. The effort of the authors for taking up such an important subject should be appreciated. I believe that the study plan chosen by the Authors is coherent and substantively correct. The manuscript was prepared correctly. The adopted methodology of statistical analysis is appropriate and precisely described. The results are presented clearly and the discussion is correct. I have no significant objections to the presented work. I would like to suggest only a few minor corrections, which I present below with the line number.

1)     It should be “SARS-CoV-2 virus”, not “COVID-19 virus”. (line 29)

2)     It should be “SARS-CoV-2”, not for example “SARS Cov-2”. (different lines)

3)     Minor linguistic, editorial and stylistic errors should be corrected.

4)     The list of references should be prepared in accordance with the editorial rules of the MDPI publishing house.

Author Response

Dear reviewer,

First of all, thank you for taking the time to review this paper. We appreciate your positive and encouraging comments. Thank you very much for your insights which greatly improved the quality of our work. Below are the responses to your reviews:

Point 1: It should be “SARS-CoV-2 virus”, not “COVID-19 virus”. (line 29)

Response 1: The term COVID-19 virus was replaced by SARS-CoV-2 virus.

Point 2: It should be “SARS-CoV-2”, not for example “SARS Cov-2”. (different lines).

Response 2: The term SARS-CoV-2 is replaced everywhere in the manuscript with "SARS-CoV-2 virus" term.

Point 3: Minor linguistic, editorial and stylistic errors should be corrected.

Response 3: The paper was submitted to the official university English teacher for revision and all changes from her side were accepted.

Point 4: The list of references should be prepared in accordance with the editorial rules of the MDPI publishing house.

Response 4: All the references are now prepared in accordance with the editorial rules of the MDPI publishing house.